# Investigation on SMT Product Defect Recognition Based on Multi-Source and Multi-Dimensional Data Reconstruction

**DOI:** 10.3390/mi13060860

**Published:** 2022-05-30

**Authors:** Jiantao Chang, Zixuan Qiao, Qibin Wang, Xianguang Kong, Yunsong Yuan

**Affiliations:** The Key Laboratory of Electronic Equipment Structure Design, Ministry of Education, Xidian University, Xi’an 710071, China; taocj@xidian.edu.cn (J.C.); qiaozx98@stu.xidian.edu.cn (Z.Q.); kongxianguang@xidian.edu.cn (X.K.); yuanyunsong777@163.com (Y.Y.)

**Keywords:** defect recognition, SMT production quality, solder paste printing, feature reconstruction

## Abstract

The recognition of defects in the solder paste printing process significantly influences the surface-mounted technology (SMT) production quality. However, defect recognition via inspection by a machine has poor accuracy, resulting in a need for the manual rechecking of many defects and a high production cost. In this study, we investigated SMT product defect recognition based on multi-source and multi-dimensional data reconstruction for the SMT production quality control process in order to address this issue. Firstly, the correlation between features and defects was enhanced by feature interaction, selection, and conversion. Then, a defect recognition model for the solder paste printing process was constructed based on feature reconstruction. Finally, the proposed model was validated on a SMT production dataset and compared with other methods. The results show that the accuracy of the proposed defect recognition model is 96.97%. Compared with four other methods, the proposed defect recognition model has higher accuracy and provides a new approach to improving the defect recognition rate in the SMT production quality control process.

## 1. Introduction

Circuit boards are core parts that enable the functionality of many electronic products. It can be said that electronic packaging technology is the driving force of SMT, and advances in SMT also promote the continuous improvement of electronic packaging technology. The performance of circuit boards is affected by the solder joints. Defects in the solder paste printing process can lead to problems with the SMT production quality. Therefore, it is essential to exercise control over the solder paste print quality during the process of manufacturing circuit boards. The solder paste printing process in surface-mounted technology (SMT) production is an essential part of the integrated circuit industry. The solder paste printing process includes main three steps, as shown in Figure 1. For the printing of solder paste, the squeegee moves horizontally in a specific direction to push the solder paste to the other side of the stencil. The solder paste has a different distribution on the solder pad owing to its flux. The distribution has a significant impact on the soldering and quality of the PCB. The recognition of defects is critical to exercising control over the solder paste print quality and ensuring an appropriate rate of qualified SMT products. Two common approaches to recognizing defects in SMT are manual visual inspection and automatic machine inspection.

In SMT production, manual visual inspection relies mainly on visual observation and the expert’s personal experience and, consequently, can be time-consuming and laborious. Moreover, some defects are invisible to the naked eye due to the solder joints being densely distributed on the PCB. Thus, manual visual inspection requires a longer time and has lower accuracy. Solder paste inspection (SPI) is most commonly used in the SMT inspection and testing process to determine whether the height, volume, area, and offset of the solder paste are within reasonable limits after the solder paste is printed on the PCB. Pre-automated optical inspection (AOI) is applied to detect missing, offset, and misplaced components before soldering. When soldering is finished, post-AOI is performed to determine the quality of the soldering and, if defects are found, the PCB needs to be rechecked and repaired manually. For exceptional cases, such as ball grid arrays, in which AOI is inappropriate, X-ray inspection is required. A combination of SPI, AOI, in-circuit tests, which are used to determine the electrical performance of components on a PCB, and automated X-ray inspection can contribute to increasing the rate of qualified SMT products and reducing rework costs. Although various machine inspection methods can be used to control the quality of each processing section with a high degree of efficiency, there remain some problems with SMT product defect recognition. Due to the high false alarm rate of the inspection equipment, its accuracy relatively poor, and existing inspection methods cannot meet the requirements. Thus, many PCBs will need to be manually re-inspected by the factory, resulting in high production costs.

In recent years, machine learning and deep learning methods have been widely applied in defect recognition due to their ability to mine datasets for correlative features and be scaled and generalized with less expertise and fewer manual operations. Machine learning methods, such as relevance vector machine [1,2], back-propagation neural networks [3,4], and convolutional neural networks (CNNs) [5,6], have been applied in defect recognition for steel surfaces. Guan et al. [6] proposed an algorithm for the recognition of defects in steel surfaces based on improved deep learning network models.

Machine vision, in combination with machine learning algorithms, has also been applied in the semiconductor industry. The authors of [7] used a robust detection method based on a vision attention mechanism and deep learning of a feature map to solve the problem of false and missed detection of casting defects in X-ray inspection. The result showed that the false detection rate and the missed detection rate for casting defects are both less than 4%. The authors of [8] used feature selection and a two-stage classifier for solder joint inspection. The result showed that the proposed scheme is not only more efficient but also increases the recognition rate. Inspired by a visual background extraction algorithm, the authors of [9] proposed an inspection method for IC solder joints with an improved ViBe algorithm that considers the defect recognition problem as an object detection problem in order to enhance the accuracy of the classification of defects. The results showed that the proposed method is universal, accurate, and easily debugged compared with other existing methods. The authors of [10] employed a method consisting of a noise filter, defect clustering using the chameleon method, and model-based pattern recognition. The authors of [11] proposed a model-based clustering algorithm for automatic spatial defect recognition in semiconductor wafers. The proposed algorithm was used to both find the defect and identify the pattern of each cluster from linear/curvilinear patterns, ellipsoidal patterns, and ring spatial patterns. Furthermore, machine learning has been extended to such applications as wood [12], the detection of cracks in tunnels [13] and oil and gas pipelines [14], and 3D metal printing [15].

Inspired by machine learning and deep learning, there has been an exhaustive investigation of defect recognition in SMT, ranging from using traditional machine learning methods to using a more modern deep learning approach. The authors of [16] used a genetic algorithm to optimize the feature extraction region and a support vector machine to classify the solder joint defect type. The authors of [17] utilized a fuzzy rule table to represent a human inspector’s criteria in order to correct any misclassification of solder joints by the neural network. The results showed that the proposed method is superior to the neural network classification method alone in terms of the accuracy of its classification. The authors of [18] proposed a dual-level defect detection algorithm, called PointNet, that extracts point cloud features from defective solder paste pattern images automatically obtained through solder paste inspection and performed defect detection at the micro-level and the macro-level for the case where two or more defects can be observed in the image. The experimental results showed that the proposed D3PointNet algorithm is robust to changes in the sparsity and size of the DSPP image, and its exact match score was 10.2% higher than that of the state-of-the-art CNN-based multi-label classification model on the DSPP image dataset. The authors of [19] used a cascaded CNN to directly perform the inspection task without low-level feature extraction and before learning to inspect SMT solder joints. Comparison experiments showed that the proposed method achieved better performance in SMT solder joint inspection than the state-of-the-art methods. Additionally, a recent study [20] attempted to inspect and classify multiple types of defects that occur in the printing stage through SPI datasets using multi-label classification for images of various sizes. The experimental results on the customized SPI dataset as well as on the VOC 2007 dataset demonstrated the excellent performance of the proposed network.

In summary, combinations of machine learning and deep learning methods have enabled remarkable achievements in defect recognition in SMT production and most of the studies are based on image data on solder paste printing obtained through machine vision or optical inspection methods for defect recognition. Moreover, the pieces of equipment involved in the production and inspection stages of solder paste printing are independent of each other, and the various types of data generated have multi-source and multi-dimensional characteristics.

However, few scholars have looked at defect recognition from the perspective of multi-source and multi-dimensional data from the solder paste printing process, resulting in a low rate of recognition of defects in SMT products. SMT innovation has become a powerful driver of the development of semiconductor and electronic manufacturing technology and has a great impact on improvements in electronic packaging technology. Therefore, in this study we investigate SMT product defect recognition based on multi-source and multi-dimensional data reconstruction. Technologies in feature reconstruction and the classification of defect types for resolving the problems described above are presented. These are also the contributions of this paper, which can be summarized as follows:(1)Feature interaction is adopted to mine for higher-order data between features. Then, the data are transformed into high-dimensional grey-image data after the RF-based feature selection method is applied to enhance the information between features;(2)A CNN is adopted for defect recognition. The CNN can effectively extract and classify the high-dimensional grey-image data obtained from the feature reconstruction.

The rest of the paper is organized as follows. Section 2 introduces the proposed methodology. Section 3 and Section 4 verify the method using an example and discuss the results of the verification. Section 5 contains our conclusions.

## 2. Proposed Method

In order to improve the accuracy of defect recognition, we propose a defect recognition method based on multi-source and multi-dimensional data reconstruction. The proposed method consists of three parts, namely solder paste printing data collection and reconstruction, feature extraction and classification, and defect recognition. First, feature reconstruction is performed on the data collected from the PCB. The interaction of the one-dimensional features is a fusion of raw data features that reflects the complex interaction between features and benefits to enhance the information about features. The CNN and a Softmax classifier are employed to construct a defect recognition model for solder paste printing defect classification. Finally, each reconstructed solder paste printing data point is classified as one of nine types of defects by the defect recognition model.

### 2.1. Feature Reconstruction

Feature reconstruction includes feature interaction, random forest (RF)-based feature selection, and feature conversion, as shown in Figure 2. First, each feature vector is multiplied by other feature vectors to achieve the interaction of features. Then, RF is applied to calculate the importance scores of features and, based on the RF result, several features with low scores are eliminated. Finally, min–max normalization is used to eliminate the influence of the dimensionality of the data on the model. The one-dimensional feature vector data are transformed into the shape of n×n.

#### 2.1.1. Feature Interaction

Feature interactions can characterize the more profound influences between defects. The feature interaction of high-dimensional data is equivalent to a higher-dimensional data space, where feature selection methods can be used to preferentially select features that are useful for classification and eliminate irrelevant features. The extension of raw data features to higher-order features can be used to take full advantage of the higher-order correlation between the raw data [21]; e.g., the relationship between the cleaning speed and the cleaning supply time characterizes the cleaning distance, and the relationship between the solder paste area and the solder paste height characterizes the solder paste volume on higher orders.

Feature interaction is performed on the original SMT training set $X=\{x_{1},x_{2},\cdots ,x_{p}\}$. All vectors are column vectors, and any two column vectors can be multiplied together to obtain the feature interaction dataset $X^{\prime }=\{x_{1,2},x_{1,3},\cdots ,x_{p-1,p}\}$. The feature interaction formula is as follows:(1)$$x_{i,j}=x_{i}\times x_{j}$$
where $x_{ij}$ represents the result of the interaction between feature $x_{i}$ and feature $x_{j}$, and the number of new feature vectors after the feature interaction is *p*(*p* − 1)/2.

#### 2.1.2. Feature Selection

Random forest (RF) is an integrated algorithm that uses a decision tree as a base learner with a bagging integration strategy. The advantages of RF for feature selection are the ability to handle high-dimensional data and the speed of computation [22]. Thus, RF is applied for the selection of new features after feature interaction. The process for calculating the random forest importance score is as follows.

The feature importance score is expressed as a VIM, and the Gini index is expressed as a GI. Assuming that there are m features $X_{1},X_{2},\cdots X_{m}$, the importance of feature $X_{j}$ at node *m*, that is, the Gini coefficient score $VIM_{j}^{Gini}$ for each feature $X_{j}$ can be calculated as:(2)$$VIM_{jm}^{Gini}=GI_{m}-GI_{l}-GI_{r}$$
where $GI_{l}$ and $GI_{r}$ denote the Gini coefficients of the two new nodes after branching, and $GI_{m}$ is expressed as:(3)$$GI_{m}=\sum_{k=1}^{\left|K\right|}\sum_{k^{\prime }\neq k}p_{mk}p_{mk^{\prime }}=1-\sum_{k=1}^{\left|K\right|}p_{mk}^{2}$$
where *K* represents the number of categories and *p_mk_* denotes the proportion of category *k* that belongs to node *m*.

Finally, all the feature importance scores are normalized to
(4)$$VIM_{j}^{Gini}=\frac{VIM_{j}^{Gini}}{\sum_{i=1}^{c}VIM_{i}^{Gini}}$$
where $\sum_{i=1}^{c}VIM_{i}^{Gini}$ is the sum of the gain of all features and $VIM_{j}^{Gini}$ is the Gini coefficient of feature $X_{j}$.

Based on the RF importance scores, features whose Gini coefficient $VIM_{j}^{Gini}$ is greater than the set feature importance threshold *d* are selected.

#### 2.1.3. Feature Conversion

After feature selection, feature conversion is performed as shown in Figure 3. It is assumed that the one-dimensional data have *q* feature vectors and a length of *l*. The data to convert are:(5)$$X=\{x_{1},x_{2},\cdots ,x_{j},\cdots ,x_{q}\}$$
where $x_{j}=\left[x_{1j},x_{2j},\cdots ,x_{ij},\cdots ,x_{lj}\right]$. Once each row of the sample dataset $y_{i}=\left[x_{i1},x_{i2},\cdots ,x_{ij},\cdots ,x_{il}\right]$ has been reshaped, the input data of the CNN can be obtained.

### 2.2. Defect Recognition Model Based on the CNN

The CNN proposed by [23] is used for image processing and pattern recognition. The CNN model includes sequentially cascaded input layers, convolutional layers, and pooling layers, a fully connected layer consisting of mutually cascaded expanded layers, and a three-layer forward neural network, wherein the forward neural network includes sequentially cascaded hidden and output layers. According to the CNN framework shown in Figure 3, the steps are as follows.

The reconstructed data are input into the CNN model, and convolution is performed at a convolution kernel to obtain the convolutional feature. The output of the convolutional layer can be calculated as
(6)$$V_{j}^{r}=f\left(\sum_{i=1}^{S}V_{i}^{r-1}*W_{ij}^{r}+b_{j}^{r}\right)$$
where *S* is the pooling filter size, * is the operator of the convolution, $V_{j}^{r}$ represents the *j*-th output feature map on the *r*-th convolution layer, $V_{i}^{r-1}$ denotes the *i*-th input feature map on the (*r* − 1)-th convolution layer, $W_{ij}^{r}$ is the convolutional kernel, which represents the *i*-th band of the *j*-th pooling filter, $b_{j}^{r}$ is the bias of the *j*-th feature map, and *f* is an activation function applied to the result, which is usually defined as a rectified linear unit (ReLU). The ReLU activation function is expressed as
(7)ReLU(x)=max(x,0)

The convolutional feature is maximally pooled through the pooling layer with a pooling filter. The specific expression for maximum pooling is
(8)$$P_{j}^{m}=\overset{p}{\underset{k=1}{\max}}(V_{j}^{(m-1)\times n+k})$$
where $P_{j}^{m}$ is the *j*-th output feature map of the *m*-th band pooling layer, and *n* and *p* are the number of sub-sampling factors and the pooling size, respectively.

The one-dimensional expansion of the second pooled feature is performed via the expansion layer to obtain the one-dimensional feature vector. The one-dimensional feature vector is used as the input into the three-layer forward neural network. For classification, the Softmax regression function was selected to be the last layer. The output of the Softmax regression function can be calculated as follows:(9)$$\mathrm{Softmax}(m_{i})=\frac{e^{m_{i}}}{\sum_{i=1}^{r}e^{m_{i}}}$$
where $m_{i}=\left[m_{1},m_{2},m_{3},\cdots ,m_{r}\right]$ is the output of the fully connected layer and *r* is the number of defect types.

The cross-entropy loss function was chosen to be the loss function used to measure the difference between the predicted values and the true values of the model’s output. During the training of the defect type classification model, the parameters need to be constantly updated so that the value of the cross-entropy loss function becomes smaller. The cross-entropy loss function is defined as
(10)$$L=-\frac{1}{N}\sum_{i=1}^{N}[y^{\left(i\right)}\log{\hat{y}}^{\left(i\right)}+(1-y^{\left(i\right)})\log(1-{\hat{y}}^{\left(i\right)})]$$
where *N* represents the total number of samples, $y^{\left(i\right)}$ represents the true label of sample x and takes a value from {0, 1}, and ${\hat{y}}^{\left(i\right)}$ represents the probability that x has the current sample label of 1. The model reaches an optimal state when the cross-entropy loss value is minimized by stochastic gradient descent [24]. The final output of the model is the predicted value of the defect type corresponding to that sample.

### 2.3. The Proposed Defect Recognition Method

The flowchart of the proposed method for SMT defect recognition based on multi-source and multi-dimensional data reconstruction is shown in Figure 4.

The proposed method has five steps: data preparation, feature reconstruction, model initialization, model training and optimization, and defect recognition.

Step 1: The dataset is divided into a training set and a test set.

Step 2: Feature interaction, random forest (RF)-based feature selection, and feature conversion are performed. Each feature vector is multiplied by other feature vectors to achieve the interaction of features. Based on the RF result, several features with low scores are eliminated. Finally, min–max normalization is used to eliminate the influence of the dimensionality of the data on the model, and the one-dimensional feature vector data are transformed into the shape of n×n.

Step 3: The CNN model includes a sequentially cascading input layer, the first convolutional layer, the first pooling layer, the second convolutional layer, the second pooling layer, and a fully connected layer consisting of mutually cascading expanded layers and a three-layer forward neural network, where the forward neural network includes sequentially cascading hidden and output layers.

Step 4: During the training of the defect recognition model, the parameters need to be constantly updated so that the value of the cross-entropy loss function becomes smaller. When the value of the cross-entropy loss function reaches its minimum or lies within an acceptable range, and a reasonable number of iterations have been run, the model reaches an optimal state.

Step 5: The optimal model is saved, and the model outputs the result of the defect recognition operation.

## 3. Experiment

### 3.1. Dataset Description

The experimental dataset used in this study is comprised of actual SMT production and inspection data, in which a variety of individual boards are involved (Table 1). The package types and package distributions contained in different individual boards are not the same. Different boards have different parameters, such as size, shape, devices, and welding temperature. Different packages may cause different types of defects. In order to increase the generalizability of the proposed model, data on multiple products were selected. The dataset is composed of process parameters, printing process status parameters, and inspection results on a total of 35,756 defect samples. A description of the physical meaning of the data is given in Table 1. The data that need to be collected during the solder paste printing stage include PCB properties, printing process parameters, printing process status parameters, and SPI results. The PCB properties include the PCB length, width, and height. The printing process parameters include squeegee speed, squeegee pressure, cleaning speed, work separation speed, work separation distance, worktable printing height offset, squeegee separation speed, and squeegee separation distance. The printing process status parameters include average pressure, minimum pressure, maximum pressure, cleaning supply time, automatic cleaning, and automatic cleaning count. The SPI results include the solder paste volume, solder paste area, and solder paste height. The real labels of each sample were tagged by SPI or a manual inspection. The manual inspection was performed after SPI and contained three detection fields, namely test-result, warn-result, and confirm-result. Test-result represents the SPI machine result, and it may be different from the manual inspection result. Warn-result represents the set value of the alarm result. Confirm-result represents the final defect as determined by the manual inspection that was performed after SPI. When the results of the three fields were consistent, the test-result data were used as the inspection result. When the results of the three fields were inconsistent, the solder paste on the pad was suspected to have a problem and, thus, needed to be manually re-checked. The confirm-result data from the manual re-check were selected as the inspection result. The processing of these inspection result labels can both help to correct defects that cannot be detected by SPI machines and reduce the number of manual re-inspections and labor costs.

A summary of the exact number of samples belonging to each defect type and the labels is shown in Table 2. A total of 70% of the raw data were used as the training set, and the remaining 30% of the raw data were used as the test set.

When the solder paste volume (SPV) value is less than 70%, the volume is too small (Small Volume) and when the SPV value is greater than 150%, the volume is too large (Large Volume). A solder paste area (SPA) that is greater than 140% is too large (Large Area), and an SPA that is less than 70% is too small (Small Area). When the solder paste has no solder paste volume (SPV), no solder paste area (SPA), or no solder paste height (SPH), a No Solder defect will occur. If the center point of the solder paste deviates by more than 20% in the positive direction of the X-axis, an X Positive Offset defect will occur. If the center point of the solder paste deviates by more than 20% in the positive direction of the Y axis, a Y Positive Offset defect will occur. If the center point of the solder paste is offset by more than 20% in the negative direction of the Y axis, a Y Negative Offset defect will occur. If the solder paste height (SPH) is greater than 170%, the height is too high (High Height).

### 3.2. Experimental Protocols

Using the grid search method [25], the values of the CNN parameters were calculated and are shown in Figure 3. Sample data of size 13 × 13 were input into the CNN model, and convolution was performed at a convolution kernel size of 5 × 5 to obtain the first convolution feature of size 13 × 13@16, where 16 represents three mapping layers, and the padding value was set to 2. Each mapping layer had an image size of 13 × 13. The first convolutional feature was then maximally pooled on the first pooling layer with a pooling filter size of 3×3 to obtain the first pooling feature of size 6 × 6@16, and the stride value was set to 2. The first pooling feature was convolved on the second convolutional layer at a convolutional kernel size of 3 × 3 to obtain a second convolutional feature of size 4 × 4@32. The second convolutional feature was maximally pooled on the second pooling layer at a pooling filter size of 3 × 3 to obtain a second pooling feature of size 2 × 2@32, the stride value was set to 2, and the padding value was set to 1. The one-dimensional expansion of the second pooled feature was performed on the expansion layer to obtain the one-dimensional feature vector. The one-dimensional feature vector was used as the input into the three-layer forward neural network. The output of the three-layer forward neural network was activated using the ReLU activation function. The final output was the predicted value of the defect type corresponding to the sample.

## 4. Results and Discussion

### 4.1. Results of CNN-Based Defect Recognition

First, the experimental data went through feature reconstruction. According to Section 2.1, each feature vector was multiplied by other feature vectors. After feature interaction, RF was employed to select 169 essential features from 210 interaction features since features with RF importance scores less than 0.00045 were deleted. The RF scores of interaction features are shown in Figure 5. Figure 5 shows that the solder paste height (SPH) influenced almost all of the main RF importance scores of feature interactions. Subsequently, min–max normalization was utilized to eliminate the influence of the dimensionality of the data. Before feature extraction, the 169 remaining interaction features were transformed into a 13 × 13 matrix.

The training data were input into the CNN model for model training after feature conversion.

The confusion matrix of the proposed method for the test set is shown in Figure 6. The data in the diagonal column represent the proportion of samples that were clustered together without being misclassified. The results indicate that a few of the samples in labels X^+^, Y^+^, and Y^−^ were mistaken for other defect types, but most of the samples under the different conditions were clustered together without being misclassified by the proposed method. The results indicate that the proposed method can effectively recognize most of the features of the nine types of defects in solder paste printing.

The validation accuracy and loss of the proposed method for feature extraction and classification are shown in Figure 7.

### 4.2. Method Comparison and Evaluation Index

Other methods were also applied on the same dataset to elucidate the effectiveness of the proposed CNN-based defect recognition method. Using the same neural network model structures, a deep neural network (DNN) [14], a deep belief net (DBN) [16], and sparse auto encoder (SAE) [17] were also introduced. The deep neural network approaches were all tested over 3000 iterations five times, and the mean classification accuracy was used for the comparison. The parameters were initialized with a weight matrix that had a mean of 0.5 and a standard deviation of 0.5. The traditional machine learning model support vector machine (SVM) [18] was introduced in order to demonstrate the advantages of the CNN in solving multi-classification problems.

(1)CNN without feature reconstruction (FR): The same training set was utilized for the CNN with the same structure without feature reconstruction. The learning rate and the batch size were 0.01 and 64, respectively, which are also the values that were used for the proposed CNN. Then, the trained model was directly applied to the test set.(2)DNN: The architecture of the standard DNN is 169-128-64-32-9. The learning rate was 0.001, and the batch size was 64.(3)DBN: The architecture of the standard DBN is 169-128-64-32-9. The learning rate and the batch size were 0.01 and 64, respectively.(4)SAE: The architecture of the standard SAE is 169-128-64-32-9. The learning rate and the batch size were 0.01 and 64, respectively.(5)SVM: The RBF kernel was applied. The penalty factor and the radius of the kernel function were 0.1 and 0.5, respectively.

After validating each model five times, the mean and standard deviation of each model were obtained. As shown in Figure 8, the model proposed in this paper has the best stability and accuracy. Table 3 lists and compares the various defect recognition methods. The weighted average recognition accuracies of the proposed CNN with feature reconstruction and without feature reconstruction are 96.97% and 82.17%, respectively. The weighted average recognition accuracies of the CNN, DNN, DBN, SAE, and SVM are 96.97%, 91.08%, 89.88%, 91.66%, and 74.84%, respectively. The results prove that, compared with DNN, DBN, SAE, and SVM, the proposed model is more effective at recognizing defects in solder paste printing.

Figure 9 displays the accuracy curves of the defect recognition results for the nine types of defects using the proposed method and the other five methods. The weighted average recognition accuracies of the CNN with feature reconstruction and without feature reconstruction range from 91.56% to 100% and from 61.43% to 99.7%, respectively. The weighted average recognition accuracies of the CNN, DNN, DBN, SAE, and SVM range from 91.56% to 100%, from 59.2% to 100%, from 81.14% to 100%, from 45.94% to 100%, and from 66.57% to 100%, respectively. The defect recognition accuracies of the other five methods are low and vary remarkably. The curve difference shown in Figure 9 and the comparison of the quantitative results shown in Table 3 illustrate the high accuracy of the proposed defect recognition method.

## 5. Conclusions

In this study, we investigated SMT product defect recognition based on multi-source and multi-dimensional data reconstruction in order to increase the low defect recognition rate and address the insufficient utilization of multi-source solder printing data in SMT production. Our conclusions are as follows.

(1)Features correlated to defects were enhanced by feature reconstruction using feature interaction, feature selection, and feature conversion. Compared with the CNN without feature reconstruction, the performance metric of recognition accuracy was improved by 14.80%.(2)Results from the experiment show that the accuracy of the proposed defect recognition model with feature reconstruction is 96.97% and the standard deviation is 1.43. Compared with four other methods, the proposed defect recognition model has higher accuracy and better stability.

Thus, the proposed defect recognition method based on multi-source and multi-dimensional data reconstruction has the potential to facilitate the widespread application of defect recognition in the SMT production quality control process. SMT, as a new type of electronic packaging technology, has penetrated every industry and every field. The defect recognition method proposed in this paper has a broad range of application prospects in such fields as electronics manufacturing and aerospace engineering.

## Figures and Tables

**Figure 1 micromachines-13-00860-f001:**
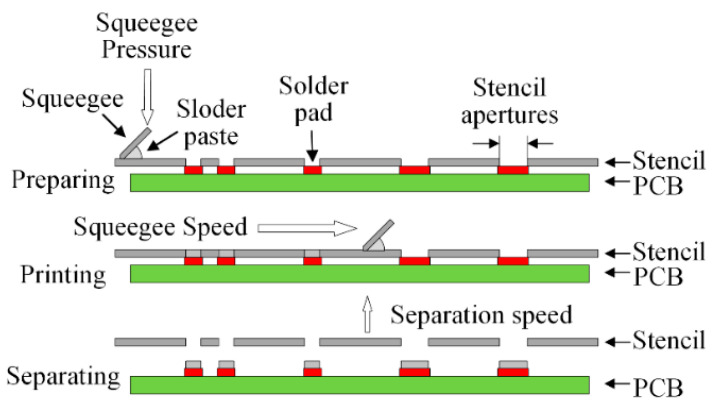
The solder paste printing process.

**Figure 2 micromachines-13-00860-f002:**
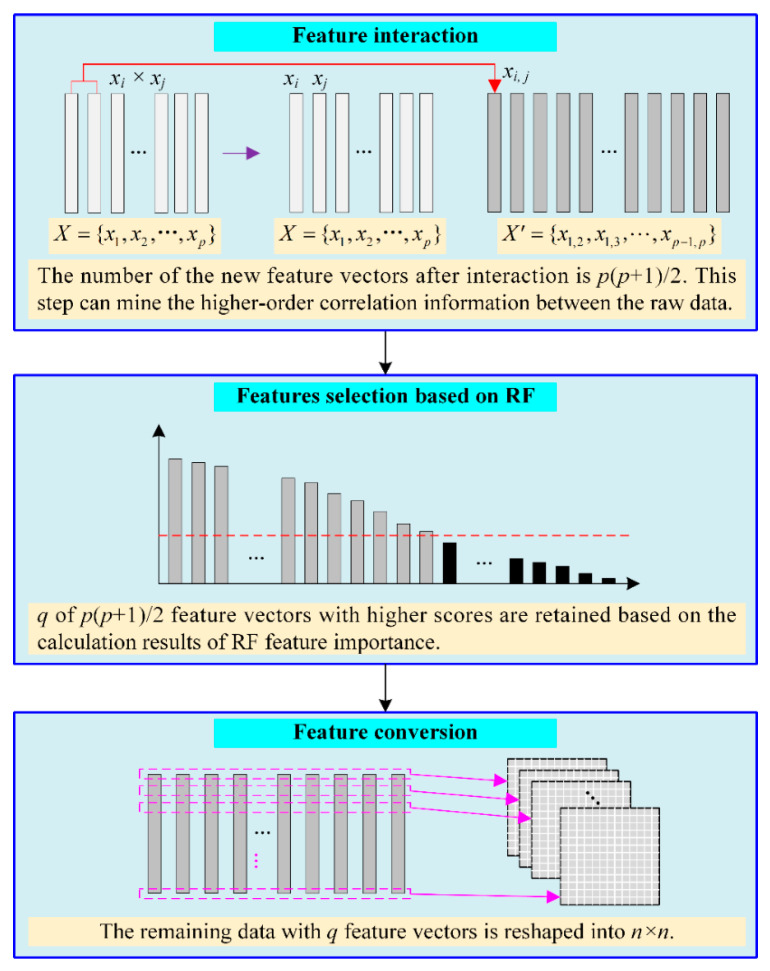
The flowchart for feature reconstruction.

**Figure 3 micromachines-13-00860-f003:**
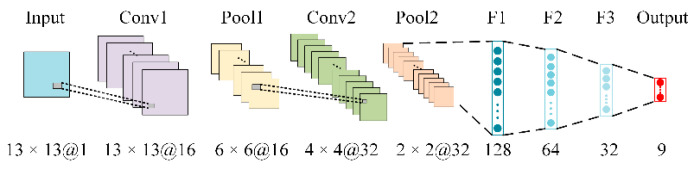
The proposed CNN framework.

**Figure 4 micromachines-13-00860-f004:**
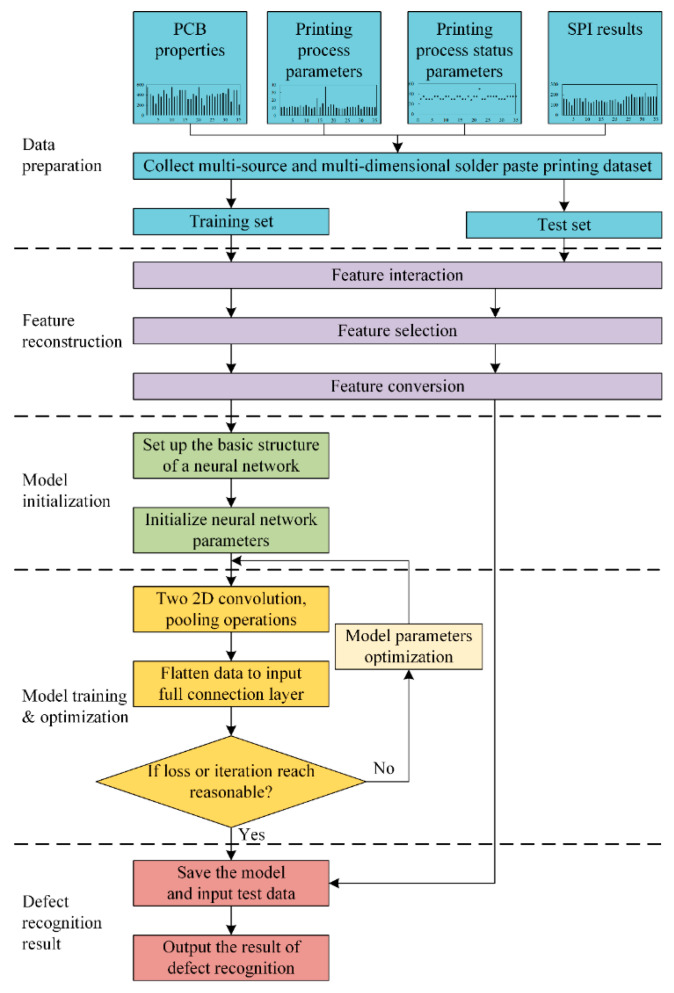
The flowchart for the proposed defect recognition method.

**Figure 5 micromachines-13-00860-f005:**
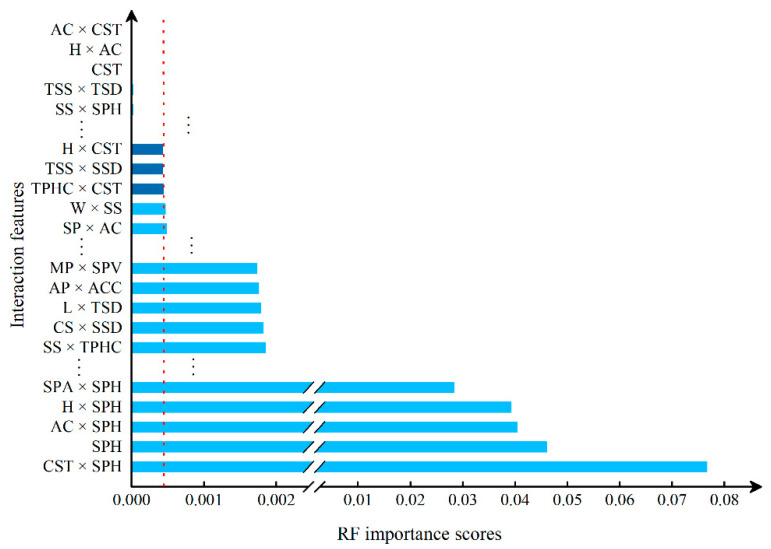
RF importance scores of interaction features.

**Figure 6 micromachines-13-00860-f006:**
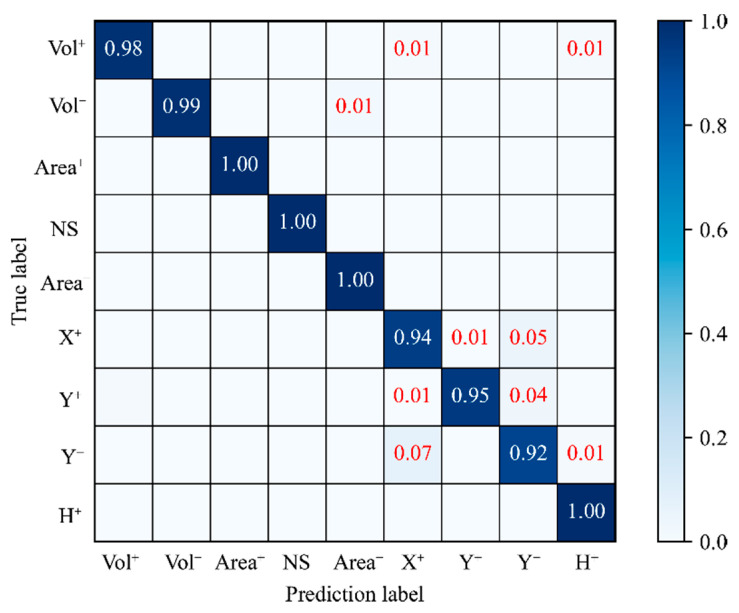
Confusion matrix of the CNN prediction results.

**Figure 7 micromachines-13-00860-f007:**
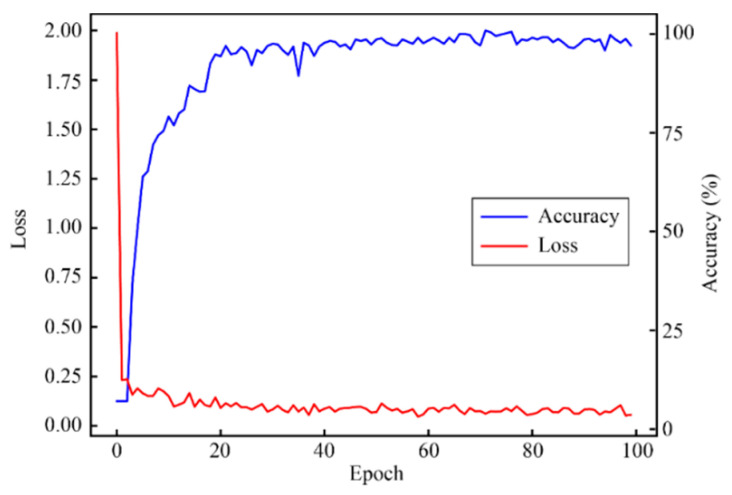
Diagram of the CNN model’s validation accuracy and loss.

**Figure 8 micromachines-13-00860-f008:**
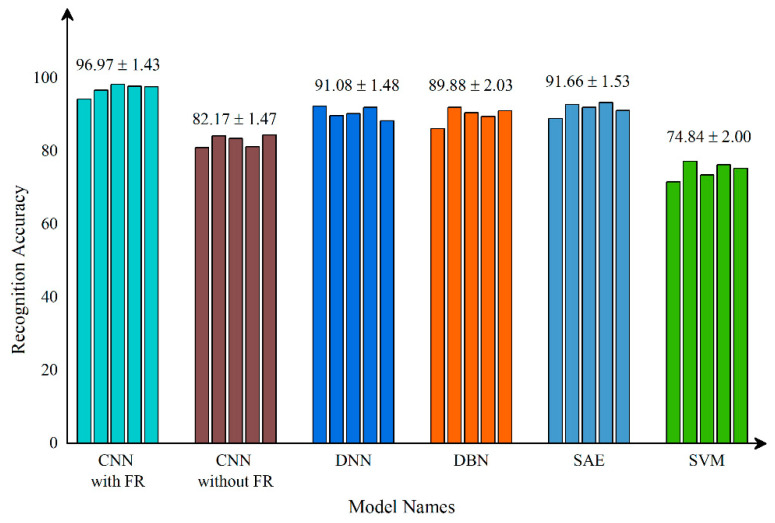
The results of different models.

**Figure 9 micromachines-13-00860-f009:**
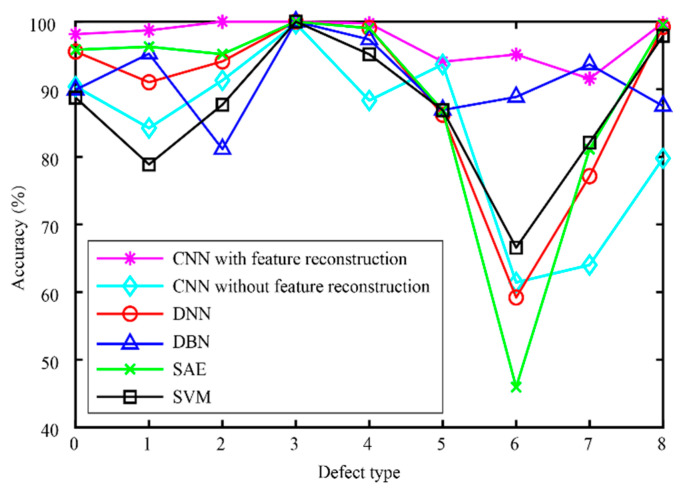
Results of different models.

**Table 1 micromachines-13-00860-t001:** Description of the physical meaning of the data.

Type	Parameter	Abbreviation	Unit	Meaning	Sample
PCB properties	PCB Length	L	mm	Length of the PCB.	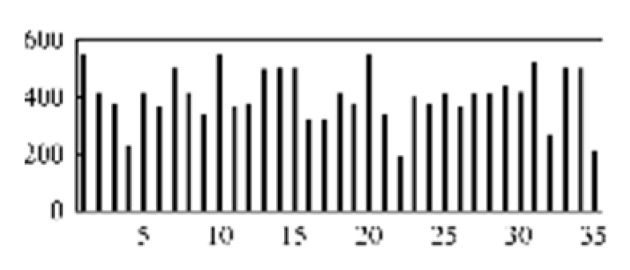
	PCB Width	W	mm	Width of the PCB.	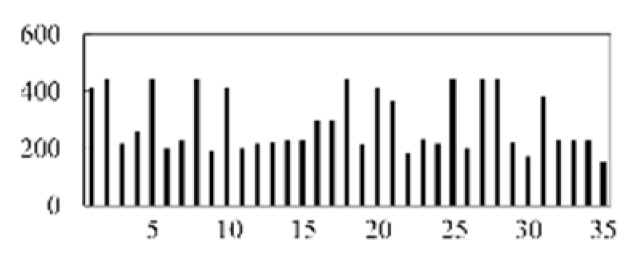
	PCB Height	H	mm	Height of the PCB.	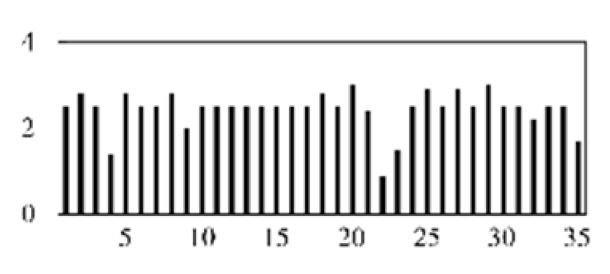
Printing process parameters	Squeegee Speed	SS	mm/s	Movement speed of the squeegee during printing.	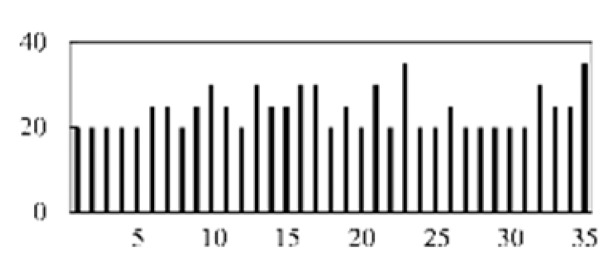
	Squeegee Pressure	SP	kg	Pressure exerted by the squeegee during printing.	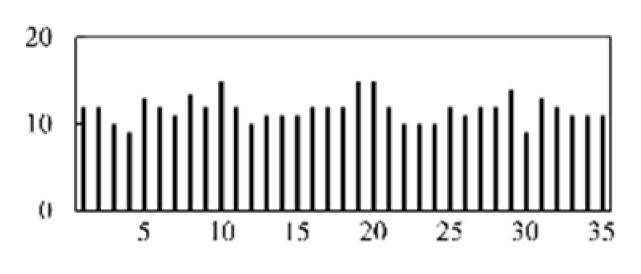
	Cleaning Speed	CS	mm/s	Stencil wiping speed during cleaning.	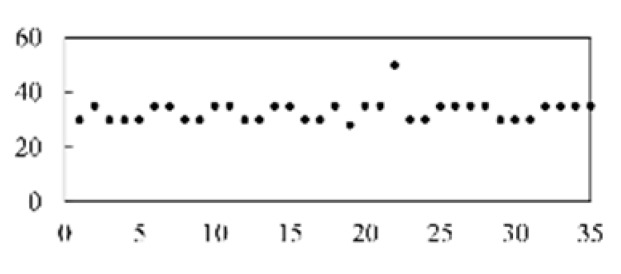
	Work Separation Speed	WSS	mm/s	Separation speed of the worktable and the stencil while the worktable is used as a platform to support the PCB, and the PCB and the stencil are separated at the end of the printing process.	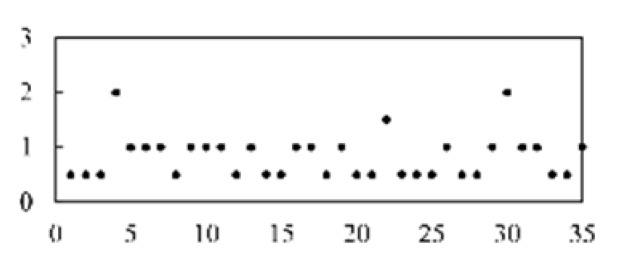
	Work Separation Distance	WSD	mm	Separation distance of the worktable and the stencil.	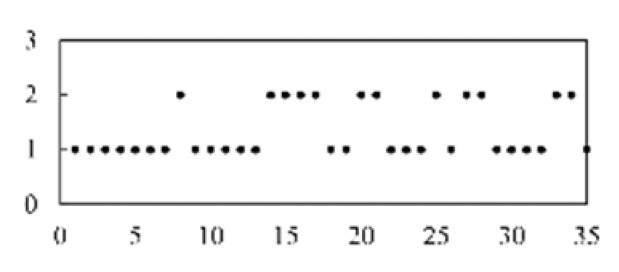
	Worktable Printing Height Offset	WPHO	mm	For different stencil thicknesses, the workbench needs different separation distances to ensure that the solder paste on the PCB is completely released from the stencil.	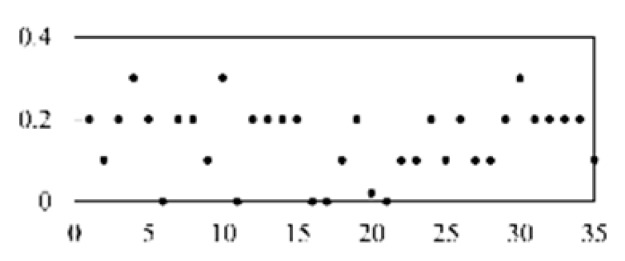
	Squeegee Separation Speed	SSS	mm/s	Separation speed of the squeegee and the stencil.	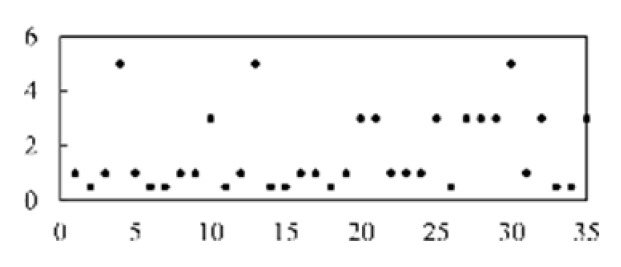
	Squeegee Separation Distance	SSD	mm	Separation distance of the squeegee and the stencil.	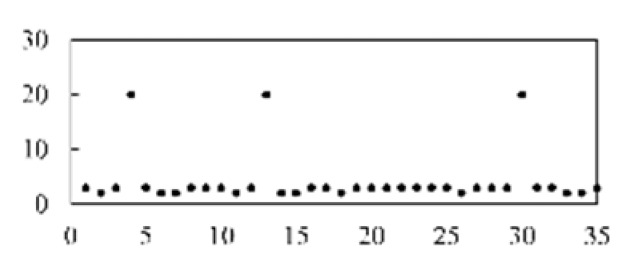
Printing process status parameters	Average Pressure	AP	kg	Average squeegee pressure.	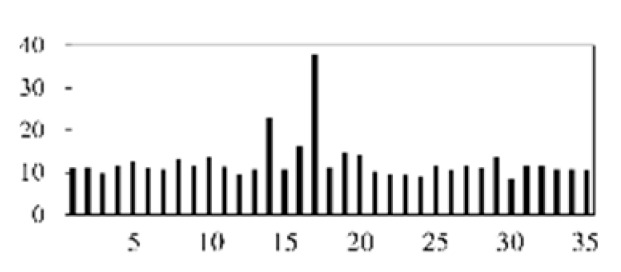
	Minimum Pressure	MinP	kg	Minimum squeegee pressure.	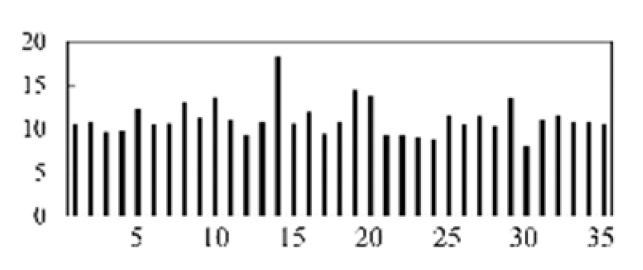
	Maximum Pressure	MaxP	kg	Maximum squeegee pressure.	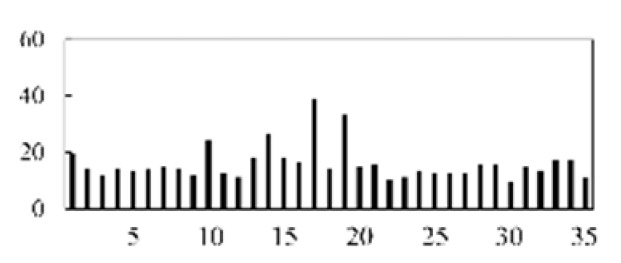
	Cleaning Supply Time	CST	s	The time taken to clean the stencil.	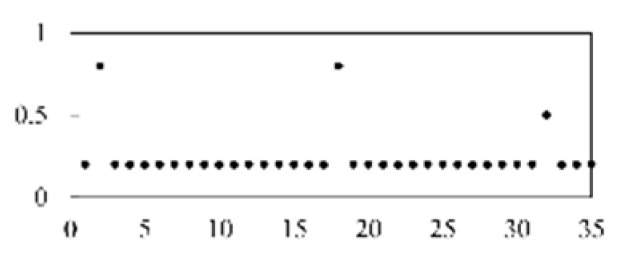
	Automatic Cleaning	AC	s	Different cleaning speeds will produce different cleaning effects on the stencil.	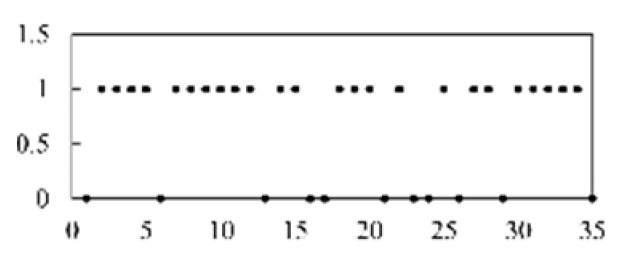
	Automatic Cleaning Count	ACC	\	Different automatic cleaning count settings will produce different cleaning effects on the stencil.	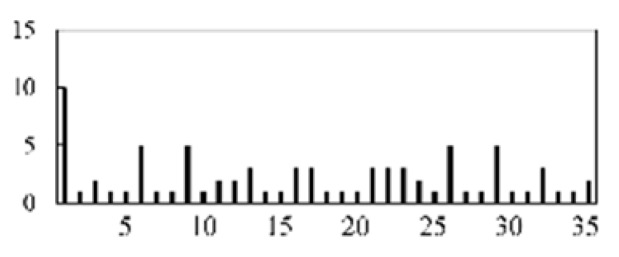
SPI results	Solder Paste Volume	SPV	\	Ratio of the measured value of solder paste volume to the theoretical value.	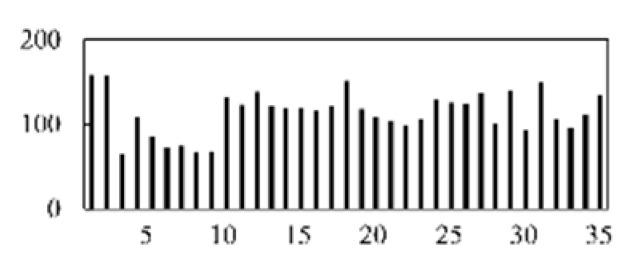
	Solder Paste Area	SPA	\	Ratio of the measured value of solder paste area to the theoretical value.	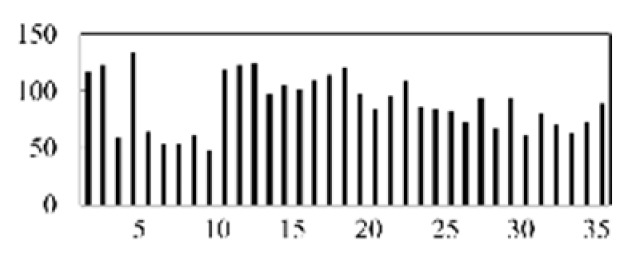
	Solder Paste Height	SPH	\	Ratio of the measured value of solder paste height to the theoretical value.	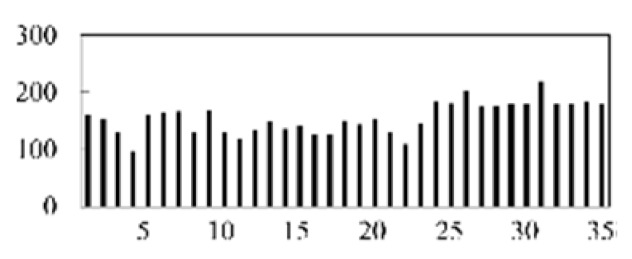

**Table 2 micromachines-13-00860-t002:** Description of the raw data.

Defect Type	Abbreviation	Label	Number of Training Samples	Number of Test Samples
Large Volume	Vol^+^	0	1389	596
Small Volume	Vol^−^	1	543	232
Large Area	Area^+^	2	531	228
No Solder	NS	3	634	271
Small Area	Area^−^	4	2561	1098
X Positive Offset	X^+^	5	5663	2427
Y Positive Offset	Y^+^	6	818	350
Y Negative Offset	Y^−^	7	3871	1659
High Height	H^+^	8	9019	3866

**Table 3 micromachines-13-00860-t003:** A comparison between the proposed model and other models.

Models	Weighted Average Recognition Accuracy (%)
CNN with FR	96.97
CNN without FR	82.17
DNN	91.08
DBN	89.88
SAE	91.66
SVM	74.84

## Data Availability

Not applicable.

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
