# Peer review of "Investigation on SMT Product Defect Recognition Based on Multi-Source and Multi-Dimensional Data Reconstruction"

_micromachines, 2022, doi:10.3390/mi13060860_

Round 1

Reviewer 1 Report

Dear authors,

I suggested to thoroughly revise the paper since the research has been conducted, according to me, very roughly...

Starting from the introduction where state of the art is completely not clear. There a lot of useless citations (line 78-90)and when it is time to describe the application of machine learning to electronic circuits (line 72-77 & line 91-102). Papers are just numbered without evidencing their results....

The lack of care is clear at line 37 where a title section is just left in the text...It should be at line 130...Methods not Proposed method...

State of the art results are resumed at line 112-114, where it is indicated a HIGH false detection and a LOW defects recognition...Are you kidding ? In process engineering these two variables are strictly related...You may suffer HIGH false detection OR LOW defect recognition...

line 141 : into is without space...

Figure 2 is a collection of the following figures...Useless and not clear at this point of the paper...

Paragraph 2.1 Feature Reconstruction that is the clou theoretical part at Line 172 has teh following equation:  P*(p+1)/2

BUT...

if you multiply the vectors and avoid same feature byproduct like showed in vector X'

it should be    p*(p-1)/2

Please motivate....

Figure 5 that is the main algorith and its following lines should go according to me to replace the uselles figure 2. Description has five steps but it misses FEATURE RECONSTRUCTION or no ?

Experimental part is pretty clear but...

In Table 2 it misses according to me some type of defects, like X negative offset and mainly "solder paste low height" that physically is different from none solder. Why weren't selected such samples ?

In the successive following lines please indictte the percentage symbol as I think it is referred to...

like 70% or 150% for SPV right ?

Section 3.3 may be also be unified in the methods section 2, in fact table 3 gives similar information to figure 4.

In the result section Figure 7 is unreadable in the tagged numbers, furthermore from figure 6 it is possible to see that solder paste height (SPH) influences all the main scores of feature interactions.

Finally the average recognition accuracy of Table 4 is calculated by simply averaging values. According to me it should be weighted according to the stochastic distribution of defects, that I suppose it may be assumed by the available samples defects distribution. 

Furthermore, If you think to give an accuracy value like 96.97% you should evaluate the uncertainty of this type of measure. Analysis

completely absent in the paper.

My best regards

Reviewer 2 Report

Nice paper, my comments:

line 11: Defect recognition of the solder paste printing process significantly influences the SMT (Surface Mounted Technology) production quality

Please explain this statement, what is the underlining evidence?

line 37: to the other side of the stencil2. Materials and Methods. Solder paste

Figure 2 & 3 -> please improve quality of the figure

line 267: The package types and package distributions contained in different single boards are not the same.

Please indicate the main differences

Page 10, SPI results. Are these the only parameters available? what about skewness?

Line 334: Before feature extraction, 169 remaining interaction features were transformed into 13×13.

13x13 matrix?

Figure 6: suggest to add legend with explanation of the abbreviations. also improve picture quality please.

Figure 7 needs a bit more explanation

Line 359: CNN without feature reconstruction -> CNN without feature reconstruction (FR)

Figure 9: why are certain defect types not accurately predicted? Please explain.

Regards,

The reviewer

Round 2

Reviewer 1 Report

Dear authors,

I recognize you put a good effort

to improve the paper and I hope

that you willingly accepted the required

improvements.

My best regards

Reviewer 2 Report

thanks for the clear update